# Learning about water resource sharing through game play

T. Ewen[1,2] and J. Seibert[1,3,4]

[1]Department of Geography, University of Zurich, Switzerland
[2]Center for Climate Systems Modeling, ETH Zurich, Switzerland
5  [3]Department of Physical Geography and Quaternary Geology, Stockholm University, Sweden
[4]Department of Earth Sciences, Uppsala University, Sweden

*Correspondence to:* T. Ewen (tracy.ewen@geo.uzh.ch)

10  **Abstract.** Games are an optimal way to teach about water resource sharing, as they allow real-world scenarios to be enacted. Both students and professionals learning about water resource management can benefit from playing games, through the process of understanding both the complexity of sharing of resources between different groups and decision outcomes. Here we address how games can be used to teach about water resource sharing, through both playing and developing water games. An evaluation of using the web-based game *Irrigania* in the classroom setting, supported by feedback from several educators 15  who have used *Irrigania* to teach about the sustainable use of water resources, and decision making, at university and high school levels, finds *Irrigania* to be an effective and easy tool to incorporate into curriculum. The development of two water games in a course for master students in geography is also presented as a way to teach and communicate about water resource sharing. Through game development, students learned soft skills, including critical thinking, problem solving, team work and time management, and overall the process was found to be an effective way to learn about water resource decision 20  outcomes. This paper concludes with a discussion of learning outcomes from both playing and developing water games.

## 1 Introduction

One of the best ways to engage students and instill enthusiasm for hydrology is to expose them to hands-on learning. Using (serious) games in the classroom can engage students, and inspire enthusiasm, while also helping to solidify formal concepts learned in standard curriculum. Learning through games has been shown to increase soft skills, such as critical thinking, 25  creative problem solving, and teamwork (Johnson et al., 2012), skills that are important for future water resource managers. When teaching hydrological concepts, and especially in the context of water resource sharing, where compromises between different interest groups need to be made and conflicts sometimes arise, games can be a good tool to enact different real-world scenarios. Learning through game play can thus be instructive in showing the complexity involved in the management of water resources, for both students and professionals alike (Douven et al., 2012; Rajabu, 2007). The active participation in 30  mock decision making, through to the outcomes of those decisions using games, also allows different learning goals, including critical thinking and problem solving, to be better realized (Wu et al., 2012).

There are several games that focus on water resources, many of which have been used and tested at various levels in educational settings. Some examples include: *Aqua Republica* (aquarepublica.com), an on-line game aimed at promoting sustainable water resource management under growing water demand and scarcity; the *World Water Game* (Deltares, 2015), where the player decides on measures to avoid water shortages in different regions of the world; and *Water: more than just a*

5 *game*, from the Swiss Federal Office for the Environment (FOEN, 2015), where the player can take different water management actions for a city and rural areas along a stream reach. These types of games focus on the player as a single actor, playing to optimize prosperity for the entire society or system. Although single actor games can have a high degree of realism by trying to simulate a real system as much as possible (Medema et al., 2016), the game can become overly complex, making it more difficult to understand and less attractive in educational settings (Jones, 2011). Although there can be a high

degree of realism in simulating the system, the idea of an individual actor is fundamentally unrealistic; in reality there are almost always many actors involved in water resource decisions. Multi-player, role-playing games, in contrast to single player games, allow different actors to interact, and are inherently more realistic as they provoke social learning and collaborative task activity (Hummel et al., 2010), and can thus be very useful in learning about water resource sharing in educational settings. Role-playing games may or may not have limited decision options that are evaluated in a quantitative

way. Examples of role-playing games with a focus on water resource sharing where players have limited decision options include board games like the *River Basin Game* and *Globalization of Water Management* (Hoekstra, 2012), that demonstrate issues related to sharing a common resource in an up- and downstream setting, incorporating the concepts of a water footprint and virtual water trade. Other role-playing games based on negotiations between different players include the Irrigation Management Game (Burton, 1989, 1994) and the River Basin Game (Magombeyi et al., 2008). In a recent review

that explores using serious games for social learning and stakeholder collaboration in transboundary watershed management, Medema et al., (2016) found that serious games, including multi-player, role-playing games, provide a promising learning platform for developing partnerships and networks, and help to increase interaction and communication between diverse stakeholder groups. Role-playing games allow players to better understand different player (stakeholder) interests and perspectives, and player dynamics, leading to specific decision outcomes. Medema et al. (2016) summarize different

characteristics of serious games that lead to success in supporting social learning and stakeholder collaborations. Among these characteristics, the degree of realism is important, but the multi-player, role-playing aspects are critical in exploring the dynamics and uncertainties involved in water resource sharing over a transboundary watershed, and ultimately lead to a better understanding of how optimal outcomes can be achieved with competing interests.

Building on the idea of better understanding multi-stakeholder decisions and how stakeholders reach an outcome (and not necessarily the optimal one), Madani (2010) suggested that game theory provides a suitable framework to study the behavior and decisions of stakeholders in water resource systems. Unlike conventional systems engineering methods which typically apply optimization methods, game theory offers a more realistic approach to studying water resource systems since people inherently have different interests, and do not always act with the best system-wide outcome in mind, which conventional

methods might assume (Madani, 2010).  Drawing on this, Seibert and Vis (2012) developed a web-based, multi-player game, which illustrates game theoretical aspects, called *Irrigania*, to teach about water resource sharing between several actors (or farmers) in educational settings. In *Irrigania* players act as farmers living in a village and decide how to irrigate their fields over several years, and are thus presented with water sharing situations with other farmers that are typical in real-world water-related conflicts. This game is simple in its rules, and there are few options for making decisions, which means that game outcomes can be more easily understood by students, making it a useful addition to a course on water resource management.

In the following, we address how effective games are in teaching about water resource sharing to different educational levels, through both game play and game development.  An evaluation of *Irrigania* in the classroom setting is first presented, supported by feedback from several educators who have used *Irrigania* for teaching about water resource conflicts at both university and high school levels.  We then discuss our experiences, together with student feedback, from a course on water games that we facilitated for masters students in geography, where students developed a board and computer game, to be used in secondary school classrooms.

## 2 *Irrigania* as a teaching tool

Since its inception, *Irrigania* (Seibert and Vis, 2012) has been used in different classroom settings and as an outreach tool, to teach about water resource sharing and to explore the role of cooperation in, and competition for the use of water as a limited common-pool resource (Seibert and Vis, 2012; Pierce and Madani, 2013; Cuadrado et al., 2014).  The game is played between villages made up of several farmers (usually 4-6 famers per village).  Each farmer has 10 fields and they can choose to irrigate the fields with a combination of rain water, river water or groundwater.  Each irrigation source has a certain cost and revenue associated with it. Rain water and river water both have a fixed cost, while the revenue for river water depends on the number of farmers using it.  For groundwater, the revenue is fixed, but the cost of groundwater increases with increasing depth to groundwater, where for $g < 8 : 20$ and for $g \geq 8 : 20 + (g–8)^2$, where g is the depth to groundwater (in arbitrary units) and dependent upon the amount of precipitation during a given year (determined by a "precipitation indicator" where a normal year = 1; a dry year = 0; and a wet year = 2) as well as the number of fields irrigated with groundwater.  In contrast, the cost of irrigating with river water is fixed at 20, but the revenue depends on the precipitation indicator (0;1;2), the number of fields irrigated with river water, and the number of farmers in the village.

The goal of the game is for each farmer to maximize his/her individual income (net of farmer revenue and costs), which to some degree requires considering the total village income. The game is usually played several times with different levels of communication and cooperation during play.  Before play the moderator (teacher) sets the length of the game, rainfall conditions and whether or not communication between farmers and/or villages occurs (making the game either cooperative

or non-cooperative), and whether users can see each other's input (information is shared).  It is recommended that several rounds be played, and the settings adjusted so that different levels of information and cooperation can be explored.  The game can also be played over several days, to give students more time to strategize and discuss results after a certain number of years have occurred, before continuing.  The student enters the "farming decisions", i.e. the number of fields irrigated with groundwater and river water, and number of rainfed fields (for a total of 10 fields), through a simple interface (Fig. 1).  The "economical status" with balance (annual income) and accumulated balance (accumulated income) of the farmer is shown, as well as the "current hydrological conditions", from which the current year's farming decisions can be based on.  The student can also see when all the farmers have made their decisions at the bottom (either "submitted" or "irrigating").  Two game scenarios are shown in Table 1: the columns (from left to right) show the game scenario (Game 1, cooperative vs. Game 2, non-cooperative); the year (1-10) for the given round; the groundwater level at the start of each year (GW level); the farming decisions taken: how many fields are irrigated with groundwater (Irrigation GW); and river water (Irrigation River); and number of rainfed fields.  The outcomes for each year follow including the income (net revenue and costs) for each year; the accumulated income for the round; and finally the accumulated income for the entire village.

After playing the game several times, patterns related to the amount of communication and information shared usually emerge (Seibert and Vis, 2012; Pierce and Madani, 2013).  In a non-cooperative setting, where no information is shared (farmers are not allowed to discuss and don't see each others input), villages typically perform worse, whereas when full cooperation occurs, and each farmer knows who the other is, there is less selfishness, more cooperation between farmers, and this high amount of cooperation usually results in a high income for the village.  This can be seen in Table 1 where two game scenarios are shown for farmer Susan from Raintown village. In Game 1 (top), a cooperative game, where players know who each farmer is, farmer Susan tends to irrigate moderately with both groundwater and river water over all years, reaches a high individual accumulated income, and her village wins with the highest accumulated village income (other villages not shown). Compared to Game 1, in Game 2 (bottom), a non-cooperative game, where players don't know who the other farmers are, farmer Susan tends to irrigate more heavily, reaches a moderate income, and has a lower overall income.  The resulting groundwater (GW) level is much lower in Game 2 at the end of the round in year 10, where GW Level = 23, compared to 19 in Game 1, reflecting the overall tendency for players to act more selfishly in the non-cooperative game setting.  Similar patterns were also found to emerge by others, e.g., Pierce and Madani (2013) who played *Irrigania* as part of a larger study to better understand decision making related to common pool resources. They showed that the most important factors to promote sustainable resource use were communication and cooperation, followed by trust, information disclosure and social learning.

When uncertainty is introduced in the weather in the *Irrigania* setting (i.e., random amount of rainfall), decisions become more difficult and differences between farmers in their risk taking also tends to emerge.  Between the different water resources, there is also learning as players improve the more they play simply by better understanding the longer term effects

of overuse in groundwater, compared to river water which, in the game, has no year-to-year memory.  In a recent study on sharing common resources among farmers in Tanzania, Lecoutere et al. (2015), showed that gender and social status were also found to play a role; during times of water scarcity, high-status women shared fairly, whereas rich and powerful men were less worried about being greedy.  Low social status (both men and women) tended to distribute water equally when it

was abundant but were more selfish when water was scarce.  These different outcomes and aspects that emerge when *Irrigania* is played with different scenarios and groups of players, make *Irrigania* a useful tool to both explore and understand the complexities of water resource sharing.

## 2.1. A survey of using *Irrigania*

To evaluate the effectiveness of *Irrigania* in teaching about water resource sharing, we carried out a survey, with an online

questionnaire sent out to users (teachers) who had registered to use *Irrigania* (since 2012; 18 in total).  We asked these users 15 questions in total and received feedback from 9 users (see Table A1, Appendix).  We asked users questions ranging from basic information on how they have used the game in their classrooms, or as an outreach tool, and how they have incorporated playing the game into their curriculum.  We then asked for details on the educational level of their class, the type of course it was used in and how many students played.  As responses, teachers have used *Irrigania* mainly at university

level, for both bachelor and graduate courses, with one exception of using it for a high school geography course with 30 students.    It has mainly been used in courses with a water resources focus (including departments of hydrology, environmental engineering, and natural resources management).   One group however, in the department of psychology, played it with students to better understand environmental decision making.  Group sizes ranged from 20 students to 110.

This was followed by more detailed questions on the specifics of play (how many times they played with the same group, and with different groups, and duration of play).  Although some groups played it only one time, most played it frequently, and some have incorporated it into their regular class curriculum.  Most groups played it once during the semester in a block of 2-4 hours, but several also played it over several weeks, with up to one full semester for play.

Following the first set of questions, we asked more targeted questions to gauge the effectiveness of *Irrigania* in engaging students (whether the game held students' interest for the duration of play and how enthusiastic students were when playing the game).   Teachers' responses depended strongly on the level of study.  For bachelor classes that used it, most said that the game held the enthusiasm of the students for the full period, and that the students were quite enthusiastic about playing it. For the graduate level courses however, many said that a 3-hour period was sufficient, since after this amount of time, the

students understood the mechanics of the game and some lost interest somewhat.  For the high school students however, they wanted more graphics and visualizations to make it more interesting, and teachers commented that this would have likely held their attention for longer periods.

Questions to evaluate the effectiveness as a teaching tool were then asked, including how well *Irrigania* taught about collaboration and conflicts with regard to shared water resources and whether there was improved understanding of shared resources like surface/river and groundwater. All teachers (regardless of level) said that *Irrigania* was moderately (4 replies) to very successful (5 replies), when asked 'how successful' (not; moderately; very) in teaching about collaboration and

conflicts with regard to shared water resources. When asked about whether they thought there was an improved understanding of shared resources like surface/river and groundwater, all answered that there was increased learning about shared water resources, but that a discussion session afterwards was key to solidifying the concepts learnt, especially for the high school and early level bachelor students.

Since *Irrigania* is based on game theory, but is also simple in its rules, it can be a good way to teach about game theoretical considerations related to water resource sharing (Seibert and Vis, 2012). As a follow-up after game play, we asked whether any interesting patterns had evolved and how much discussion the teachers incorporated into the process of playing the game (e.g., whether they had discussions on the topics before and/or after play). We then asked a few questions related to game theory including whether game theoretical considerations related to water resource sharing were discussed (before and/or

after playing) and whether *Irrigania* was successful in teaching students (or other players) about the tragedy of the commons. Almost all teachers discussed game theoretical considerations related to water resource sharing briefly before play, but also in a final discussion after play, and this also helped to solidify learning concepts related to game theory. Almost all teachers also found that students understood, by the end of the session play, that cooperative behavior and communication were both key to succeeding. All teachers said that *Irrigania* was successful in teaching students about the

tragedy of the commons and supporting discussion of these concepts (all answered 'yes' to this).

Additional questions were asked on whether the teacher had used other educational games, and differences they found in teaching aspects in these games compared to *Irrigania*. Four teachers used other games in the classrooms, and all said that in comparison, *Irrigania* was very easy to use and required little preparation before using it in the class, which made it

appealing. In a final question, we asked for general feedback that teachers thought would be useful for evaluating *Irrigania* as an innovative tool for learning about water resource sharing and suggestions for improving the game. Several suggestions were given, e.g., for younger students (high school) it was suggested that it should be more game-like and visually engaging. University level students however seemed to find it engaging enough, but also suggested that a spatial interface be developed where villages could be represented visually. It was also suggested that more game settings would make it more interesting,

allowing students to explore more scenarios and play longer, for e.g., by setting different amounts of water from different sources and having rewards or punishments for level of sharing. Two teachers recommended that a more flexible groundwater level evaluation be implemented by allowing the game to be played with different amounts of available water to start. Another commented that allowing the results to easily be exported would be an advantage for follow-up discussion and analysis of game play.

Overall, the feedback from the survey was positive, and all teachers felt that *Irrigania* was a good tool for teaching about both shared water resources, and game theory. The results highlight that the use of *Irrigania* for different levels of teaching is quite different, and that it seems to be best suited to higher bachelor level to master level courses where students were the

most engaged, it held their interest for longer, and teachers had less comments for improvements for these groups.

Additional analysis was carried out considering user data collected since July 2013, when user histories began to be saved; this excluded data collected during our own use of *Irrigana*. This data included how often users played *Irrigania* (number of games played), how long their rounds were (average game length), and over what period of time they played. The number of

games played varied from only one game (users 8,9) to 26 games played (user 10), with most users playing games with 10 years (the default setting), although user 10 played consistently shorter games, with an average of five years. For the game length, many users played over one day, but users 1 and 12 played over a 2 month period, and user 10 (with 26 games played), playing over the full period (July 2013 - present). This agrees with some of the user feedback from the online questionnaire, where many teachers had used it once during the semester in a block of 2-4 hours, and several also played it

over several weeks, with up to one full semester for play.

## 3 Developing water games in the classroom

An 'Integrative Project' course within the master's program at the Department of Geography at the University of Zurich, is a six credit point course, corresponding to 180 working hours for the students, running over two semesters. This course has the aim of putting theory learned in the classroom into practice, and is led by different teachers or research groups within the

geography department each year. In the "Integrative Project" course on "Water Games" (fall term 2014 and spring term 2015) five students, four female and one male, from the MSc program in geography participated. All students had German as a mother language and the class was taught partly in German and partly in English. In the following, we first present the course as well as the design and development of two games by students that participated in the course, followed by an evaluation of learning outcomes from the course.

A first goal of the course was for the students to carry out a survey of existing water-related games, including both computer and board games. These games were then played and both positive and negative aspects of each game were discussed, followed by an analysis of what makes a good game. Students also had a couple of lectures, with one on project management followed by two lectures on game theory, given by invited game theory experts, introducing students to game

theory (which *Irrigania* is based on). The second part of the course focused on the development of their own games, first through brainstorming ideas for new games, and then forming groups. The students then developed two different games: a board game, *Wiapuna* (Figure 2, and a computer based game, *Habitat Ganges* (Figure 3) over a period of 6 months. Game development began with initial 'idea boards' (Figure 4) where students brainstormed possible game ideas, discussing aspects

of each in class, and further in working group sessions, to narrow down their ideas. Most ideas built upon already existing games that the students had reviewed and played in the first part of the course. The games were then developed over three months of group work with students organizing their own group time together (including summer). During game development, students also tested (played) the games with a couple of smaller groups of their intended target audiences, to

get feedback and make improvements. In a final three hour class, the games were played by the students in the class and other geographers in the department. Overall, the players enjoyed the games and comments for improvements or changes were discussed amongst the players.

*Wiapuna:* Wiapuna was developed as a multiplayer board game (Figure 2) for both family play or play in schools or as an

outreach tool, for ages 10 and older. It is based on the topic of water resource scarcity, and could be incorporated into regular geography curriculum to supplement and enhance regular lectures. In Wiapuna, players build and develop settlements around four central wells (Figure 5), where water is supplied by buying water pipes, and shared between neighbors using the same well. Natural resources (copper, gravel, wood and food, Figure 5 right) are used to buy infrastructure. Water supply through wells is slowly depleted as more and larger houses are built around each well. New

efficiency measures need to be implemented to reduce the amount of water use (e.g., through buying drip irrigation, harvesting rainwater for agriculture, and increasing efficiency in household appliances). An element of uncertainty is introduced into the game with natural events that include global and regional heavy rainfall, water poisoning, floods, droughts, tornados or storms. The board design is based on the well known *Settlers of Catan* board game, where players are also awarded points as their settlements grow, and like *Settlers*, is won by the first player to reach a certain number of points.

Game play is approximately 70-100 min long, and thus could be incorporated into the regular curriculum, where several sessions could be devoted to game play.

*Habitat Ganges:* Habitat Ganges is an online game (Figure 3) about the sustainable use and sharing of water resources along the Ganges. This game is aimed at German speaking geography students in secondary schools, ideally for groups of 16-24 students. Time needed is approximately 90 mins, which could be played in a classroom where 2 x 45 min sessions could be

planned for play (approx. 15 rounds). The focus of the game is on the development of sustainable water use for communities (the cities of Kanpur, Varanasi, Calcutta and the district of Chamoli), and the consequences for the river, the communities relying on it, and the environment, caused by poor river management. Students developed the game based on the sustainability triangle, described by Heins (1994), as a way to show that sustainability needs to be approached by considering ecological, economical, and social aspects equally and all together, in an integrative way. They applied this to the idea of

river management and the interaction between upstream and downstream use. The overall objective of the game is to create a sustainable river environment between the different communities (played in teams), with each community's action affecting the others, as in the case of a real river with upstream-downstream consequences for each community. The game is played by buying and trading resources (with the different resources shown in the field; Figure 5; Table 2), in an attempt to optimize the economy, life quality, and water quality of the Ganges (Table 2 "Effects"), starting with a certain budget. The game is

won by achieving the highest overall score from these three indicators, while also taking into account the total population and remaining budget.

### 3.1 Evaluation of "Water Games" course

Based on feedback we received after the course from the students, one of the main comments that most of the students had about this course, was that the time (two full semesters), was not enough to get introduced to different games, get into groups, and finally develop, test and produce their own games. In the end, the rush to complete a final project, and actually produce a game (especially the board game which required a lot of technical expertise to produce) that could be played during the final session (and used later on as a teaching or outreach tool), meant that the game testing phase was very limited. Since the course was really aimed at getting students to apply theory to practice, there is a goal to produce a product at the end that can be used for either teaching or as a communication tool. This problem in time management likely resulted out of a combination of this (not having much experience in turning theory into a practical product in their studies), and having difficulty getting started with the project (deciding on a group and idea and getting going). The latter could have been improved by giving students more time at the beginning of the course to discuss ideas. The introductory sessions/lectures could have been shorter, and possibly more direction while developing ideas and forming the groups given.

Students commented that the lecture on game theory was maybe the least useful part of the course, although they found it interesting, several said that what they learned in the lectures was too theoretical and not useful for them to immediately apply in their game development. Following the lectures, the next part of the course, where students reviewed existing games, worked rather well, and the students all gave positive input about this part and said it was critical for them in developing their own game ideas. This was also clear in the development of the final games, since both of the games were based on existing games that they had reviewed during this part of the course. After this, when students were given time to get into groups, discuss ideas and get down to work, proved to be challenging – some students had quite strong ideas about how they wanted to proceed, and what type of game they wanted to develop (based on their skills, interests and review of what makes a good game), without wanting to discuss too much with other students. This was however to be a group activity, and reaching a consensus was rather important for the game development to get started. In the end it was decided that the two games would be developed, and that one of the students would contribute to both groups. Once this decision was made, game development went reasonably smoothly, and students spent many hours discussing and testing the intricacies and complexities of water resources sharing. In each step of game development, all the possibilities resulting from of each player's next move had to be evaluated, and through this process, many scenarios were thought through to the final outcome. This process meant that students learned about water resources sharing in great detail and that soft skills learning, including critical thinking, problem solving and team work, was reinforced. Several students who didn't have a background in either physical geography or hydrology also participated in the course and although their learning curve for the material was very steep, had an excellent grasp of the topic after having developed their games.

The overall impression of the course from students was that they had put a lot of work into the course (for the given number of credit points received) – the group project was intense, requiring them to meet and work together frequently. The deadline for the final games to be submitted was also extended into summer and the next fall semester, but they nevertheless scrambled to get the games finished over the summer holiday. As mentioned, this course was meant to emphasize practical aspects of what students learn during their master's curriculum, and students found the transition from theory to practice to be a more challenging step. Although they also had a course on project management, most of them felt that they couldn't apply the information learnt to their actual project. Indeed, working through the theory of project management, is not likely useful without a concrete project to apply those theories to. This lecture could have maybe come later in the course, after they had formed groups, and finalized their project ideas, and then finally apply some of the project management principles to their planning. Given these minor glitches, the students were quite satisfied with having taken the course, and produced their games, and it was definitely a very new (learning) experience for everyone. A next step is to now to get others to play the games, either incorporating the games into teaching curriculum for the age appropriate levels, or possibly during hydrology/water focused outreach events as a communication and teaching tool.

## 4 Discussion and Conclusions

In this paper, we have presented a short evaluation of how both playing games and developing games can be effective ways to learn and communicate about water resource sharing. Using *Irrigania*, a multi-player, web-based game, we presented results from a survey carried out to evaluate the effectiveness of its use in the classroom to teach about water resource sharing. Our survey showed that *Irrigania* is an effective tool for learning about: i) water resource sharing, and that both cooperation and communication are key factors for sustainable water use; ii) different shared resources including surface/river and groundwater and differences between them; and iii) tragedy of the commons and support discussion of these somewhat theoretical and sometimes difficult concepts for students to grasp. Overall, teachers found *Irrigania* to be an effective and also easy tool to incorporate into curriculum, ideally for upper level bachelor to master level students, studying either water resources or decision making.

Learning activated through both playing and developing serious games in the classroom can provide crucial skills for future professionals to solve complex water resource problems. The complex learning through game play and game development emphasizes problem-solving, communication and collaboration, and critical reflection on wicked problems (Hummel et al., 2010), of which water resource management is one. In a review of learning outcomes of playing serious games, Wouters et al., (2009) found that serious game play improve the acquisition of knowledge and cognitive skills, and that they seem to be promising in accomplishing attitudinal change, likely an important aspect for future water resource professionals as they transition from an educational setting to the workplace, bringing new perspectives with them. In a study on using serious games in acquiring water resource management skills, Hummel et al. (2010) found that the aspect of collaboration within

serious games (in the classroom setting) can improve learning about certain problem situations applied in the workplace, according to new modes of more active and experiential learning. The focus on cooperation and communication in *Irrigania,* through its multi-player character and simple game set-up, where communication between farmers is decided before game play, thus also likely lead to improved learning of water resource sharing concepts.

An evaluation of a course on developing water games, based on our experience and student feedback, found that designing and developing their own water games was a positive learning experience for students, although they found it somewhat difficult putting theory into practice to produce their final games. Developing their own games was an active learning exercise, emphasizing what Ruben (1999) describes as "social, collaborative, and peer based" learning. During game
development, students had to think through and discuss the intricacies and complexity of water resource sharing, as they enacted players' moves and water resource outcomes, and then had to reevaluate game variables. Through this process, fundamental learning about water resources took place, emphasizing soft skills, including critical thinking, problem solving, collaborative (team) learning and time management. Several studies that have looked at the effects of collaborative learning in serious game development (Corrigan et al., 2015; Prensky, 2003; Mansour and El-Said, 2008), found that the development
of serious games (within the workplace (Corrigan et al., 2015)) play a role in fostering the development and improvement of various soft skills, such as communication, collaboration or negotiation and enhance overall collaborative learning, similar to learning outcomes from playing serious games. Corrigan et al. (2015) further suggest that "we are at the beginning of a fundamental shift in the way both learning and working is happening in organisations", and that these novel, active learning tools, including both playing and developing serious games, can add a critical collaborative dimension to decision making
that cannot be learned otherwise. Our course was a first step in testing serious game development in the classroom and further insight into the learning outcomes as well as carry-on effects into the workplace would be an interesting research question which could shed light on whether just playing games (emphasizing the fun factor), might be enough to achieve similar learning effects as the full process of game development.

## Acknowledgements

We thank the students in GEO401: Water games, Marc Vic for helping to create and continue managing *Irrigania*, Sandra Pool for helping with pedagogical aspects of the "Water Games" course, and the research group H2K for playing games with us. Our water games can be found at: http://www.geo.uzh.ch/en/units/h2k/services/water-games/

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

# Irrigania

## Year 4

**Farming decisions:**

| | |
|---|---|
| Irrigation (Groundwater) | 3 |
| Irrigation (River) | 2 |
| Rainfed | 5 |

**Economical status:**

| | |
|---|---|
| Balance this year | 180 |
| Accumulated balance | 640 |

**Current hydrological conditions:**

| | |
|---|---|
| Depth to groundwater | 5 |
| Costs for irrigation using groundwater (per field, last year) | 20 |

Year **3** was a **normal** year
There was enough riverwater to irrigate all fields sufficiently.

Submit

**Villages and Users:**

Raintown
- Jacob (Submitted) (1 - 1 - 8)
- Hans (Irrigating) (2 - 5 - 3)

Watervillage
- Michael (Irrigating) (2 - 4 - 4)
- Susan (Irrigating) (5 - 1 - 4)
- Otto (Irrigating) (2 - 0 - 8)

Figure 1: The student (farmer) web-interface during a game of *Irrigania* showing the "Farming decisions" taken for Year 4, the "Economical status" based on Years 1-3, and "Current hydrological conditions" to base the current year's decisions on.

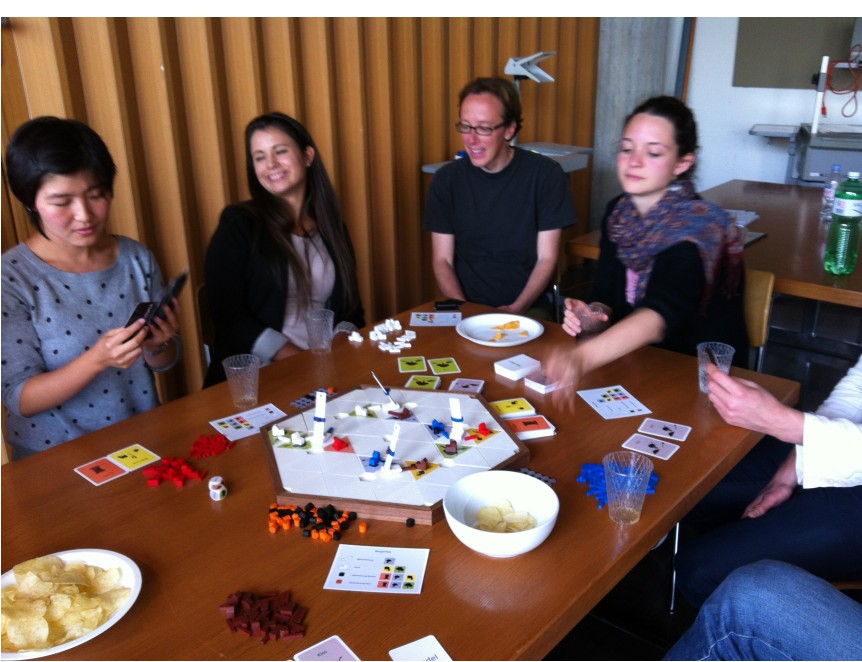

Figure 2: Playing the board game *Wiapuna* in the final class.

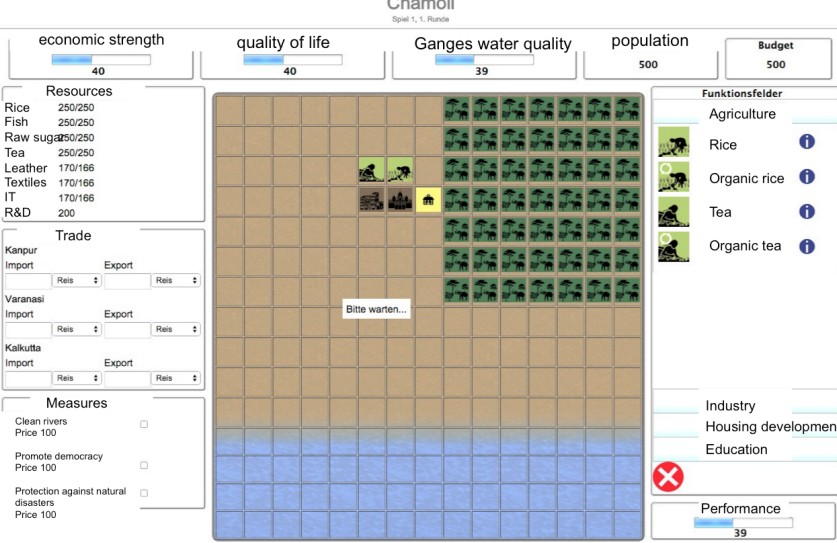

Figure 3: A screen-shot of *Habitat Ganges* - more than just a game (Lebensraum Ganges - Mehr als ein Spiel). Shown is the game interface for the district of Chamoli, translated from the German. Note that the resources here can be related to those shown in the resource price list in Table 2.

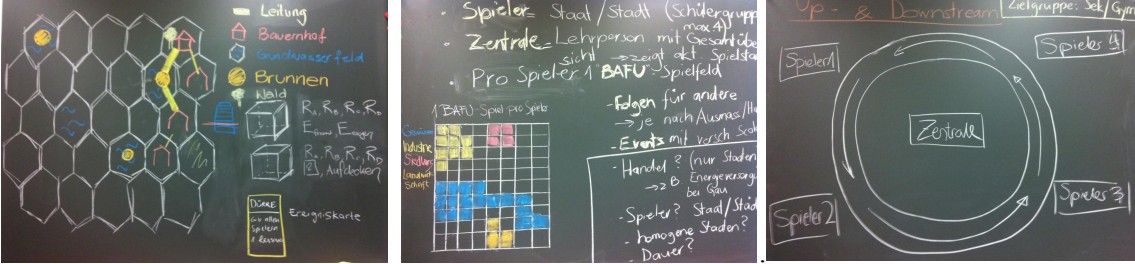

Figure 4: Initial stages of game development with idea boards. Board 1 (left): shows a hypothetical game board with options for introducing a pipeline (Leitung); farmyard (Bauernhof); groundwater source (Grundwasser Feld); well (Brunnen), forest (Wald); drought (Dürre) -> event card' (Ereigniskarte). Board 2 (middle): game board development based on the FOEN, 2015 game. Board 3 (right): hypothetical game idea for computer game based on the idea of upstream downstream river use and influence on each player (Spieler).

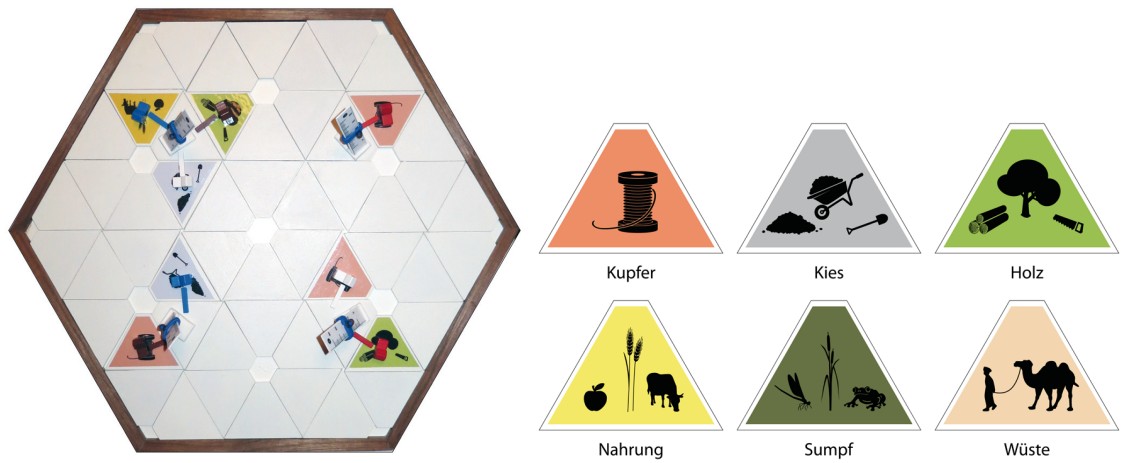

Figure 5: Board set-up for Wiapuna centered around four wells (left). Settlements are developed on different land use tiles (right), corresponding to the natural resource cards (copper (Kupfer), gravel (Kies), wood (Holz), food (Nahrung), marsh (Sumpf) and desert (Wüste)) that are used to buy infrastructure and energy efficiency measures.

| Game 1: Cooperative | Year | GW Level | Irrigation GW | Irrigation River | Rainfed | Income | Accum. Income | Accum Income Village |
|---|---|---|---|---|---|---|---|---|
| Village: Raintown | 1 | 7.25 | 2 | 3 | 5 | 525.00 | 525.00 | |
| Farmer: Susan | 2 | 9.25 | 2 | 2 | 6 | 453.54 | 978.54 | |
| | 3 | 11.25 | 3 | 3 | 4 | 548.31 | 1526.85 | |
| | 4 | 11.75 | 3 | 2 | 5 | 442.81 | 1969.67 | |
| | 5 | 12.25 | 2 | 1 | 7 | 378.88 | 2348.54 | |
| | 6 | 13.5 | 3 | 4 | 3 | 530.92 | 2879.46 | |
| | 7 | 15.25 | 2 | 1 | 7 | 309.88 | 3189.33 | |
| | 8 | 17.75 | 3 | 2 | 5 | 239.81 | 3429.15 | |
| | 9 | 18.25 | 2 | 3 | 5 | 294.88 | 3724.02 | |
| | 10 | 19 | 3 | 4 | 3 | 272.00 | 3996.02 | **16745.02** |
| | | | | | | | | |
| Game 2: Non-cooperative | Year | GW Level | Irrigation GW | Irrigation River | Rainfed | Income | Accum. Income | Accum. Income Village |
| Village: Raintown | 1 | 6.5 | 4 | 4 | 2 | 650.00 | 650.00 | |
| Farmer: Susan | 2 | 8 | 4 | 4 | 2 | 663.33 | 1313.33 | |
| | 3 | 10.25 | 4 | 4 | 2 | 669.75 | 1983.08 | |
| | 4 | 12.25 | 5 | 4 | 1 | 601.35 | 2584.44 | |
| | 5 | 14.75 | 5 | 4 | 1 | 517.19 | 3101.63 | |
| | 6 | 17 | 6 | 6 | -2 | 424.00 | 3525.63 | |

| | 7 | 19.75 | 4 | 5 | 1 | 142.75 | 3668.38 | |
| | 8 | 23 | 5 | 5 | 0 | -325.00 | 3343.38 | |
| | 9 | 22.5 | 1 | 5 | 4 | 236.42 | 3579.79 | |
| | 10 | 23 | 1 | 5 | 4 | 271.67 | 3851.46 | **11012.46** |

Table 1: Two Irrigania game scenarios played with international students during a course at CABI (Centre for Agriculture and Biosciences International), Delemont, Switzerland: Game 1 (top), a cooperative game, and Game 2 (bottom), a non-cooperative game, for farmer Susan in Raintown village. Farmer Susan tends to irrigate more heavily in Game 2, acting
more selfishly, ending up with a lower individual income and a lower accumulated income for her village, as compared to Game 1 where the other farmers in Raintown are known to her.

| Resource | Price/year | Yield/year | | Effects | | |
| | | Resource | Budget | Economy | Quality of life | Ganges water quality |
|---|---|---|---|---|---|---|
| Agriculture/Fisheries | | | | | | |
| Tee plantation | 60 | 30 | 20 | + | 0 | - |
| Rice field | 60 | 30 | 20 | +++ | 0 | -- |
| Sugar cane plantation | 60 | 30 | 20 | +++ | 0 | -- |
| Fishery | 60 | 30 | 20 | +++ | 0 | -- |
| Industry | | | | | | |
| Textile factory | 80 | 50 | 60 | +++++ | + | --- |
| Leather factory | 80 | 50 | 60 | +++++ | 0 | ----- |
| IT firm | 90 | 60 | 70 | ++++++ | + | --- |

Table 2: Each community in Habitat Ganges is given a sheet of paper indicating the list of prices for each resource (in arbitrary monetary units) together with the qualitative outcome (+/-) for each of the indicators (economy, life quality, and water quality) needed to win the game (here only "Agriculture/Fisheries and Industry" are shown for Calcutta resource prices, as an example).

Appendix: Table A1: Irrigania Survey. The Irrigania survey questions (16, left column) sent out to 18 Irrigania users. A total of 9 users responded. Responses are shown for each question, and comments when given.

| Irrigania survey: use in the classroom and for outreach events | Responses | Comments |
|---|---|---|
| **1.** Have you used Irrigania in a classroom setting? [Yes/No] | Yes, 8<br>No, 1 | |
| **2.** If yes, what educational level was it used for? | High school, 1<br>University, bachelor level, 4<br>University, graduate level (masters/PhD), 3 | |
| **3.** What was the name of your course and what department/institute is it in? | Risk Analysis, School of Environmental Engineering, (Greece);<br>Geography, Secondary 2 (high school; US);<br>Geography, Oregon State University (US);<br>Natural Resources Management and Integrated Water Resources Management (Italy);<br>Engineering Systems Design (Singapore);<br>Behavioral psychology, Dept. Psychology (US);<br>Hydrology, Geography;<br>Water resources, Environmental Engineering | not all responded; country provided in brackets where given |
| **4.** If you have used Irrigania to teach about water concepts outside of a classroom setting, please let us know what kind of event it was, e.g., an outreach event or during a meeting. | | no responses |
| If you've played Irrigania with students and/or other groups of players, please answer the questions below: | | |
| **5.** How many students (or other players) played Irrigania? | group size (number of replies)<br>1-10 (2)<br>11-20 (2)<br>21-50 (2)<br>50-80 (1)<br>> 80 (1) | |
| **6a.** How many times have you played Irrigania with the same group of students (or other players)? | 0 (2)<br>1 (2)<br>3 (2)<br>3 games/same day (1)<br>> 10 (1) | |
| **6b.** How many times have you played Irrigania with different groups of students (or other players)? | 0 (2)<br>1 (2)<br>2 (3)<br>> 5 (1) | |
| **7.** How long did the students (or other players) play Irrigania? | 1 hour (1)<br>2 hours (2)<br>3 hours (2)<br>over one week (2)<br>over one semester (1) | |
| **8.** Did the game hold their enthusiasm for this length of time, or could the session have been shorter/longer? | longer (3)<br>- yes, the students were excited by Irrigania and wanted to play longer<br>- yes, ideally it should be played for more than 2 hours, e.g., 3-4 hours.<br><br>shorter (2)<br>- It is a wonderful game but the lack of visuals and graphics made it a little less engaging for the students, who are easily distracted and bored with things.<br>- The session could have been a bit shorter as the students' enthusiasm decreased after they understood the mechanisms of the game. | |

| | |
|---|---|
| **9.** How interested/enthusiastic were the students (or other players) about the game? | very interested (3)<br>very interested initially, but lost interest after ~1 hour (2)<br>very interested in the game competition (2)<br>very interested in setting up different strategies and testing them, e.g.,<br>cooperative vs. non-cooperative (1) |
| **10.** How well in general did Irrigania teach about collaboration and conflicts with regard to shared water resources? [Very/Moderately/Not very successful] | very successful (5)<br>moderately successful (3)<br>not very successful (0) |
| **11.** Do you think there was improved understanding of shared resources like surface/river and groundwater? | yes (8)<br>Yes, but most didn't get that far.<br>Yes, but it is important to recall and consolidate these concepts in a debrief session. |
| **12.** Did you notice any interesting patterns that evolved when playing the game in a class? | - Cooperative behavior was improved among players<br>- Yes. In the first rounds students were taking decisions a bit randomly. After this (testing phase), decisions started to be more rational and related to the objectives of the game. |
| **13.** Did you discuss game theoretical considerations related to water resource sharing? Before or after playing (each round)? | Before (3)<br>After (2)<br>Before and after (3) |
| **14.** Do you think Irrigania was successful in teaching students (or other players) about the tragedy of the commons? | yes (6)<br>yes, more or less (2) |
| **15.** Have you used other educational games? If so, which ones? What differences did you find in teaching aspects compared to irrigania? | No (4)<br>Catchment Detox<br>http://www.abc.net.au/science/catchmentdetox/files/home.htm |
| Please give any other information that might be useful in evaluating Irrigania as an innovative tool for learning about water resource sharing. | - Allow for more flexible groundwater levels<br>- It has the potential to be a powerful educational tool, but it might need to be more engaging and more game-like.<br>- It will be very useful for older children/young adults<br>- Allow for more game settings, e.g., allow for different amounts of available water, rewards<br>- Improve the user experience, include a nice interface with spatial representation of the villages.<br>- Would be great if the results could be directly exported in some formats (e.g., Excel). |

