# Peer review of "Learning about water resource sharing through game play"

_Hydrology and Earth System Sciences, 2016_

## Referee Comment (RC1) · M. Jones (Referee) · 11 Mar 2016

General Comments

This paper presents a useful initial analysis of the learning benefits that may be derived from game play of water resource sharing.

The paper documents a range of games available and how one particular game, Irrigana, appears to be developing as a learning platform. The sample size on which the analysis is based is small and further analysis would be useful in future to support the conclusions drawn. To provide more context for those unfamiliar with Irrigana, it would be useful to provide example input scenarios, decisions and outcomes, preferably visualised, to help the reader appreciate more fully the value and potential of the game. More importantly, to support the evaluation of game play benefits, it would be useful to

include the survey questionnaire used. Similarly, including the survey results as a table would be useful to clarify the description of the evaluation.

The paper also presents an evaluation of the benefits to learning about water resource sharing derived from developing games. This element of the paper needs to reviewed; the paper would be improved if it identified the specific points of student learning on water resource sharing that have been derived from developing new games.

Specific Comments

Section 2, Irrigania as a teaching tool Page 2, Line 25 - The text notes that Irrigana assumes "...cost of groundwater increases with increasing depth to groundwater." It would be useful to understand the basis on which this depth increases, presumably the amount and duration of pumping.  In this context, it would also be useful to understand how any interactions between groundwater  rivers are represented.  These points may be covered by Siebert  Vis, but a brief comment here would help appreciate the conceptual hydrological system represented in Irrigana and therefore, to what scenarios the game can be applied.

Page 2, Line 26 - The text states maximising income is the goal of the game, while previously revenue is mentioned.  To improve clarity it would worth being specific that the income is net of farmer costs, if this is the case, and differs from revenue.

Section 2.1, A survey of using Irrigania Although there were few respondents to the survey, it would be useful to understand where all of the Irrigana users were based, whether they responded or not. This would provide extra information on the geographical spread or restricted distribution of responses and so the international penetration of Irrigana as a learning tool.

It is important to include the survey questionnaire used to underpin the results presnted and conclusions drawn.  Although this may take up a significant amount of space, it would be useful as the questioning is multi-stage and not simple to follow with a textonly description.

It would help as well to present the survey results as a table, including the number of respondents at each stage of the questioning. This should help make the results more accessible to the reader and enable an appreciation of the confidence in the conclusions that have been drawn. This would also help the explanation of results on page 4 line 17-18 and on page 5 line 21-28.

The use of brackets rather than commas can be a matter of personal preference, but in Section 2.1 this results in parts of the text being awkward to read. A particular example to address is on Page 4, Line 16 where nested brackets are used, but are incomplete. To aid the reader, I'd suggest that this and other sentences be reworded to allow many of the brackets to be removed.

Section 3 Page 6, Line 12, reference to Figure 1 - Suggest spiltting Figure 1 left and Figure 1 right into separate figures. This would help enable an explanation/translation of the German text  labelling to be included. Unfortunately, the text is inaccessible for those unfamiliar with German.

Page 6, Line 13, reference to Figure 2 - It is useful to have Figure 2 included to illustrate game development, but referencing of Figure 2 left (Line 21) and Figure 2 right (Page 7, Line 1) needs to be clarified. For example, it's unclear if there should be a reference to Figure 2 middle and if so, it's very unclear what Figure 2 left actually illustrates and what it adds to the documentation of game development.

Page 6, Line 25, reference to Figure 3 - Including an explanation/translation of the German labelling would help understanding of the Wiapuna game.

Page 7, Line 12, reference to Table 1 - Column headers include Price/year and Yield/year, but the units for price and yield are no specified.  If the intention is that they are dimensionless and illustrative in the context of the game, then this needs to be clarified.

Section 3.1, Evaluation of learning outcomes The key messages from game development seem to relate mainly to insufficient time, planning challenges and need for re-timetabling of other course modules. This is interesting, but the evaluation would benefit from documenting more substance on the value and benefits to learning about water resource sharing derived from the games developed.

In this context, the conclusions on the game development state that the "students had to think through the intricacies and complexity of water resource sharing, as they thought through players' moves and water resource outcomes", but there is no detail on what these intricacies and complexity were. This is in contrast to the learning experiences from using Irrigana noted in Section 2.1, which at least highlights that the learning has been that "cooperative behavior and communication were both key to succeeding". It would improve the paper's contribution if it identified the specific points of learning on water resource sharing that have been derived from developing the games.

Technical Corrections

Page 1, Line 23 - Reference to Johnson, 2012 should either be Johnson et al. or the paper is missing from the reference list.

Page 2, Line 20 - To improve clarity, suggest rewording as follows, ".... role of cooperation in, and competition for the use of water as a limited common-pool resource"

Page 3, Line 15 - Reference should read Lecoutere et al. (2015)

Page 5, Line 21 - Rewording suggested as follows "Additional analysis was carried out considering user data collected since July 2013, when user histories began to be saved; this excluded data collected during our own use of Irrigana. This was done to further analyse how ......"

Page 7, Line 7 - Insert "a" as follows, ".....Heins (1994), as a way to show.."

---

## Referee Comment (RC2) · Anonymous Referee #2 · 13 Jun 2016

The paper presents an interesting and innovative learning tool to understand resource management and use. The manuscript begins with a review of a range of games available but no critical input is provided as to what the limitations are of the reviewed examples and why the new game presented is different. No important contribution is put forward as to 'what is the new aspect this new game provides that hasn't been provided already by the other games?' the review is therefore short of analytical substance and would require more work in order to identify gaps in the current knowledge and use of these types of games and how the new game presented is different and ultimately better? The manuscript lacks a proper discussion of the implications of the use and results of the game once it has been played. The manuscript should include a section on implications for management, and a discussion as to how these results are relevant in the real world? How can managers/practitioners learn from this new

knowledge and advance groundwater management? What should be the lessons and messages to take home with that? The scope of the manuscript is therefore limited to the 'classroom' and doesn't do much to advance 'further and wider knowledge' on groundwater management. The manuscript therefore lacks 'vision' and would require re-thinking as to the real lessons to be drawn from the work that is presented. Further details on the data used (as suggested by the other reviewer) in the form of a table with descriptive statistics of the results would be interesting to have.

---

## Author Comment (AC1) · 8 Jul 2016

RC1 comment 1: The paper documents a range of games available and how one particular game, Irrigana, appears to be developing as a learning platform. The sample size on which the analysis is based is small and further analysis would be useful in future to support the conclusions drawn.

AC1 reply: We agree that the sample size is small if one looks on the number of teachers who shared their experiences with Irrgania. However, each of these teachers represents ca 10-50 students, which means that the experiences are based on many people all together. For the future we plan to continue collecting feedback from Irrigania users in order to increase the sample size, and support the conclusions drawn in this paper, but obviously a significant increase of the sample size will take quite some time

(years)

RC1 comment 2: To provide more context for those unfamiliar with Irrigana, it would be useful to provide example input scenarios, decisions and outcomes, preferably visualised, to help the reader appreciate more fully the value and potential of the game.

AC1 reply: We had discussed whether to put in specific game scenarios from Irrigania, but decided that including a reference to the original paper by Seibert and Vis was likely sufficient. Given this helpful comment, we will include two scenarios with game decisions and outcomes in the revised manuscript to help the reader better understand how the game is played, and potential scenario outcomes.

RC1 comment 3: More importantly, to support the evaluation of game play benefits, it would be useful to include the survey questionnaire used. Similarly, including the survey results as a table would be useful to clarify the description of the evaluation.

AC1 reply: We weren't sure on the format for including both the questions and responses from the survey and agree that this could be improved. As suggested by the reviewer, we will include a table in the revised manuscript with the survey questions and summary of survey results.

RC1 comment 4: The paper also presents an evaluation of the benefits to learning about water resource sharing derived from developing games. This element of the paper needs to reviewed; the paper would be improved if it identified the specific points of student learning on water resource sharing that have been derived from developing new games.

AC1 reply: We thank you for this helpful comment. In the paper, we briefly discuss (in the "Discussion and Conclusions" section) what types of learning the students gained from their game development, including: soft skills, critical thinking, problem solving, team work and time management. We agree with the reviewer that these points could be further discussed in the paper. In the revised manuscript we will elaborate on these

skills and highlight the most important learning outcomes, based on both the feedback from the students and our assessment of the course. Although we touched on learning outcomes in other games, we will now tie in the feedback we received with findings from the literature we cited, to further support the discussion.

RC1 specific comment 5: Section 2, Irrigania as a teaching tool Page 2, Line 25 - The text notes that Irrigana assumes "...cost of groundwater increases with increasing depth to groundwater." It would be useful to understand the basis on which this depth increases, presumably the amount and duration of pumping. In this context, it would also be useful to understand how any interactions between groundwater rivers are represented. These points may be covered by Siebert Vis, but a brief comment here would help appreciate the conceptual hydrological system represented in Irrigana and therefore, to what scenarios the game can be applied.

AC1 reply: The cost per field of irrigating with groundwater is given by: for $g < 8$ : 20 and for $g \geq 8$ : $20 + (g–8)2$, where g is the depth to groundwater (in arbitrary units of the order of m) and dependent upon the amount of precipitation during a given year (determined by a "precipitation indicator" where a normal year = 1; a dry year = 0; and a wet year = 2) as well as the number of fields irrigated with groundwater. In contrast, the cost of irrigating with river water is fixed at 20, but the revenue depends on the precipitation indicator(0;1;2), the number of fields irrigated with river water, and the number of farmers in the village. This is described in detail in Seibert and Vis, but we will now include a short description of this so the reader can better understand outcomes of different scenarios between different resources used. RC1 specific comment 6: Page 2, Line 26 - The text states maximising income is the goal of the game, while previously revenue is mentioned. To improve clarity it would worth being specific that the income is net of farmer costs, if this is the case, and differs from revenue. AC1 reply: This is a very good point, and see that we've used the two words interchangeably, but in fact only revenue is considered in Irrigania. We will adjust the text so it is correct. RC1 specific comment 7: Section 2.1, A survey of using Irrigania: Although

there were few respondents to the survey, it would be useful to understand where all of the Irrigana users were based, whether they responded or not. This would provide extra information on the geographical spread or restricted distribution of responses and so the international penetration of Irrigana as a learning tool.

AC1 reply: This is a very good point, we have this information from the survey and can include it.

RC1 specific comment 8: It is important to include the survey questionnaire used to underpin the results presnted and conclusions drawn. Although this may take up a significant amount of space, it would be useful as the questioning is multi-stage and not simple to follow with a text- only description. It would help as well to present the survey results as a table, including the number of respondents at each stage of the questioning. This should help make the results more accessible to the reader and enable an appreciation of the confidence in the conclusions that have been drawn. This would also help the explanation of results on page 4 line 17-18 and on page 5 line 21-28.

AC1 reply: We agree, and as written above in response to the reviewer's general comment on this, we will include this information as a table so it's clearer.

RC1 specific comment 9: The use of brackets rather than commas can be a matter of personal preference, but in Section 2.1 this results in parts of the text being awkward to read. A particular example to address is on Page 4, Line 16 where nested brackets are used, but are incomplete. To aid the reader, I'd suggest that this and other sentences be reworded to allow many of the brackets to be removed.

AC1 reply: Thank you for this comment, we will reword these sentences and remove the brackets.

RC1 specific comment 10: Section 3 Page 6, Line 12, reference to Figure 1 - Suggest spiltting Figure 1 left and Figure 1 right into separate figures. This would help enable

an explanation/translation of the German text labelling to be included. Unfortunately, the text is inaccessible for those unfamiliar with German.

AC1 reply: We agree and will split these figures and include the translation of the German text in the caption for (the current) Figure 1 left.

RC1 specific comment 11: Page 6, Line 13, reference to Figure 2 - It is useful to have Figure 2 included to illustrate game development, but referencing of Figure 2 left (Line 21) and Figure 2 right (Page 7, Line 1) needs to be clarified. For example, it's unclear if there should be a reference to Figure 2 middle and if so, it's very unclear what Figure 2 left actually illustrates and what it adds to the documentation of game development.

AC1 reply: Thank you for noticing this. The figure numbers are incorrect in the text and the references should be to Figure 1 left and right and not Figure 2 left and right. We will correct this in the text.

RC1 specific comment 12: Page 6, Line 25, reference to Figure 3 - Including an explanation/translation of the German labelling would help understanding of the Wiapuna game.

AC1 reply: We agree and will put a translation of the German text in the caption.

RC1 specific comment 13: Page 7, Line 12, reference to Table 1 - Column headers include Price/year and Yield/year, but the units for price and yield are no specified. If the intention is that they are dimensionless and illustrative in the context of the game, then this needs to be clarified.

AC1 reply: These values are given in arbitrary units of money, thank you for pointing this out. We will add this to the text (and Table 1 caption).

RC1 specific comment 14: Section 3.1, Evaluation of learning outcomes The key messages from game development seem to relate mainly to insufficient time, planning challenges and need for re-timetabling of other course modules. This is interesting, but the evaluation would benefit from documenting more substance on the value and benefits to learning about water resource sharing derived from the games developed. In this context, the conclusions on the game development state that the "students had to think through the intricacies and complexity of water resource sharing, as they thought through players' moves and water resource outcomes", but there is no detail on what these intricacies and complexity were. This is in contrast to the learning experiences from using Irrigana noted in Section 2.1, which at least highlights that the learning has been that "cooperative behavior and communication were both key to succeeding".

AC1 reply: Thank you for this helpful comment. In addition to the points mentioned above (reply to comment 4), we will include more text and explanation on the learning outcomes (now outlined in the discussion), and relate these more clearly to the "intracacies and complexity" that we mentioned in Section 3.1. We also thank you for the point about comparing this to Irrigania, and think it would be worth adding some text to link the learning experiences from both Irrigania and the student developed games. This was overlooked and including this will help to strengthen the paper.

RC1 specific comment 15: It would improve the paper's contribution if it identified the specific points of learning on water resource sharing that have been derived from developing the games.

AC1 reply: Yes, we agree and will add more discussion on this as outlined in the comments above (replies to comments 4, 14).

RC1 technical correction 16: Page 1, Line 23 - Reference to Johnson, 2012 should either be Johnson et al. or the paper is missing from the reference list.

AC1 reply: Yes, this is incorrect and should refer to Johnson et al. (2012). We will correct this in the text.

RC1 technical correction 17: Page 2, Line 20 - To improve clarity, suggest rewording as follows, ".... role of cooperation in, and competition for the use of water as a limited common-pool resource"

AC1 reply: Thank you, we will reword this.

RC1 technical correction 18: Page 3, Line 15 - Reference should read Lecoutere et al. (2015)

AC1 reply: Thank you for noticing this, we will correct it.

RC1 technical correction 19: Page 5, Line 21 - Rewording suggested as follows "Additional analysis was carried out considering user data collected since July 2013, when user histories began to be saved; this excluded data collected during our own use of Irrigana. This was done to further analyse how ......"

AC1 reply: We will reword following the suggestion for clarity.

RC1 technical correction 20: Page 7, Line 7 - Insert "a" as follows, ".....Heins (1994), as a way to show.."

AC1 reply: Thank you for noticing, we will include this.

We thank Michael Jones for his very careful review of our paper and his valuable comments. We think that by including his more substantial comments on the Irrigania questionnaire, and further clarifying the learning outcomes, the manuscript will be substantially improved.

––––––––––––––––––––––––

---

## Author Comment (AC2) · 8 Jul 2016

RC2 comments: The paper presents an interesting and innovative learning tool to understand resource management and use. The manuscript begins with a review of a range of games available but no critical input is provided as to what the limitations are of the reviewed examples and why the new game presented is different. No important contribution is put forward as to 'what is the new aspect this new game provides that hasn't been provided already by the other games?' the review is therefore short of analytical substance and would require more work in order to identify gaps in the current knowledge and use of these types of games and how the new game presented is different and ultimately better?

AC2 reply: We thank the reviewer for this helpful comment. We will include more

literature in the introduction to help identify the gaps in the current literature regarding other types of games that are currently used for teaching about water resource sharing. This will help to better compare Irrigania with the other games, and allow the strengths of Irrigania to be better identified. We agree that this could be substantially improved with a more comprehensive literature review and help to highlight what makes Irrigania novel.

RC2 comment: The manuscript lacks a proper discussion of the implications of the use and results of the game once it has been played.

AC2 reply: This is a very helpful comment. In the text we wrote that "cooperative behavior and communication were both key to succeeding", which was actually based on feedback from teachers who had discussed the outcomes with their classes after the students played. In some cases, students played on more than one occasion, and usually students notice that these factors (cooperative behavior and communication) are key to succeeding and so approach the next game with this in mind (and thus usually change their strategy based on this outcome). We will try to make these "implications of the use and results of the game" more clear in the text, and try to link these ideas better.

RC2 comment: The manuscript should include a section on implications for management, and a discussion as to how these results are relevant in the real world?

AC2 reply: Thank you for this helpful comment. We can include a short section on implications for management and relevancy in the real world, based on the feedback and outcomes of the game as played in the classroom setting.

RC2 comment: How can managers/practitioners learn from this new knowledge and advance groundwater management? What should be the lessons and messages to take home with that?

AC2 reply: Although we refer to the fact that Irrigania may be useful for water resource

managers, we don't currently have any feedback from this user group to (we feel) support any further comments on this. We can however comment on this in the text as regards to student learning in the classroom, and how this learning in the classroom setting may be relevant for these students in further careers in water management, and address these questions in this context.

RC2 comment: The scope of the manuscript is therefore limited to the 'classroom' and doesn't do much to advance 'further and wider knowledge' on groundwater management. The manuscript therefore lacks 'vision' and would require re-thinking as to the real lessons to be drawn from the work that is presented.

AC2 reply: Although the scope of our manuscript is indeed clearly focused on "classroom" aspects, we believe that learning about groudwater management starts in the classroom – it is in the classroom where future water resource managers are trained, and think that this learning does get carried forward. It would be nice to have some feedback/data from water resource managers and practitioners to further identify real lessons. Although our data is currently limited to teaching about water resource sharing in the classroom, we strongly believe that there is value in this information to better improve our educational programs and training in water resource management. We do however agree with this comment insofar as we could try to connect our findings with how they might feed into real world lessons, and add a sentence on this in the discussion.

RC2 comment: Further details on the data used (as suggested by the other reviewer) in the form of a table with descriptive statistics of the results would be interesting to have.

AC2 reply: Thank you for this comment. We agree and will improve this, also according to RC1's comments (and outlined in replies to RC1 comments 3, 8). We hope this will help to clarify and better explain the results, and improve the readability of the manuscript.

We would like to thank reviewer #2 for all the helpful comments and questions. Although we would like to be able to better address the questions related to "vison" and real lessons in water resource management, our current study (and data) is limited to the classroom. We will however certainly address these points in the discussion, as they are relevant and would be very interesting to pursue as a follow-up to this study.

---

## Editor Comment (EC1) · 21 Jul 2016

Dear Tracy and Jan,

many thanks for your detailed and positive responses to the referees' comments. Both reviewers were broadly favourable about the paper but provided very specific suggestions for amendments to improve it, and I note in your replies that you have largely accepted those.

For me, the over-arching issues are:

(1) the early part of the paper needs a more critical appraisal of the literature on game play in this area, making clear in what ways Irrigania improves or advances this.

(2) The later part of the paper needs more emphasis on the wider implications of the

game play - taking the lessons from the classroom and examining their significance for professional water management. This revision accepts that this study was classroom-focused, but simply aims to make clear in the Discussion/Conclusions the broader utility of the approach.

I look forward to seeing a revised paper uploaded so that I can progress this further. I'd appreciate it if you could provide a supplement outline highlighting those changes that you have made in relation to the reviewer's comments.

many thanks,

Iain

---

## Author Response (AR1)

**Reply to reviewer 1 "Learning about water resource sharing through game play" by T. Ewen and J. Seibert**

**Reply to RC1: M. Jones (Referee)**
michael.jones@thameswater.co.uk

RC1 comment 1: The paper documents a range of games available and how one particular game, Irrigana, appears to be developing as a learning platform. The sample size on which the analysis is based is small and further analysis would be useful in future to support the conclusions drawn.

AC1 reply: Thank you for this helpful comment. We agree and plan to continue collecting feedback from Irrigania users in the future in order to increase the sample size, and support the conclusions drawn in this paper.

RC1 comment 2: To provide more context for those unfamiliar with Irrigana, it would be useful to provide example input scenarios, decisions and outcomes, preferably visualised, to help the reader appreciate more fully the value and potential of the game.

AC1 reply: We had discussed whether to put in specific game scenarios from Irrigania, but decided that including a reference to the original paper by Seibert and Vis was likely sufficient. Given this helpful comment, we have included two scenarios with game decisions and outcomes in the revised manuscript (Table 1) to help the reader better understand how the game is played, and potential scenario outcomes.

RC1 comment 3: More importantly, to support the evaluation of game play benefits, it would be useful to include the survey questionnaire used. Similarly, including the survey results as a table would be useful to clarify the description of the evaluation.

AC1 reply: We weren't sure on the format for including both the questions and responses from the survey and agree that this could be improved. As suggested by the reviewer, we have included Table A1 (Appendix) in the revised manuscript with the survey questions and summary of survey results.

RC1 comment 4: The paper also presents an evaluation of the benefits to learning about water resource sharing derived from developing games. This element of the paper needs to reviewed; the paper would be improved if it identified the specific points of student learning on water resource sharing that have been derived from developing new games.

AC1 reply: We thank you for this helpful comment. In the paper, we briefly discuss (in the "Discussion and Conclusions" section) what types of learning the students gained from their game development, including: soft skills, critical thinking, problem solving, team work and time management. We agree with the

reviewer that these points could be further discussed in the paper. We have now added more text into the Discussion and Conclusions section where learning outcomes in both game play and development are discussed and relevant literature cited.

RC1 specific comment 5: Section 2, Irrigania as a teaching tool Page 2, Line 25 - The text notes that Irrigana assumes "...cost of groundwater increases with increasing depth to groundwater." It would be useful to understand the basis on which this depth increases, presumably the amount and duration of pumping. In this context, it would also be useful to understand how any interactions between groundwater rivers are represented. These points may be covered by Siebert Vis, but a brief comment here would help appreciate the conceptual hydrological system represented in Irrigana and therefore, to what scenarios the game can be applied.

AC1 reply: The cost per field of irrigating with groundwater is given by: $g < 8 : 20$ $g \geq 8 : 20 + (g{-}8)^2$, where g is the depth to groundwater (in arbitrary units) and dependent upon the amount of precipitation during a given year (determined by a "precipitation indicator" where a normal year = 1; a dry year = 0; and a wet year = 2) as well as the number of fields irrigated with groundwater. In contrast, the cost of irrigating with river water is fixed at 20, but the revenue depends on the precipitation indicator(0;1;2), the number of fields irrigated with river water, and the number of farmers in the village. This is described in detail in Seibert and Vis, and we have now include a short description of this in Section 2 (page 3, lines 23-27) so the reader can better understand outcomes of different scenarios between different resources used.

RC1 specific comment 6: Page 2, Line 26 - The text states maximising income is the goal of the game, while previously revenue is mentioned. To improve clarity it would worth being specific that the income is net of farmer costs, if this is the case, and differs from revenue.

AC1 reply: This is a very good point, and see that we've used the two words interchangeably. To improve clarity we have now specified that income is net of farmer costs, and differs from revenue (page 3, line 29).

RC1 specific comment 7:
Section 2.1, A survey of using Irrigania: Although there were few respondents to the survey, it would be useful to understand where all of the Irrigana users were based, whether they responded or not. This would provide extra information on the geographical spread or restricted distribution of responses and so the international penetration of Irrigana as a learning tool.

AC1 reply: We have added this information into a summary of the survey in Table A1 (Appendix).

RC1 specific comment 8:
It is important to include the survey questionnaire used to underpin the results presnted and conclusions drawn. Although this may take up a significant amount

of space, it would be useful as the questioning is multi-stage and not simple to follow with a text- only description. It would help as well to present the survey results as a table, including the number of respondents at each stage of the questioning. This should help make the results more accessible to the reader and enable an appreciation of the confidence in the conclusions that have been drawn. This would also help the explanation of results on
page 4 line 17-18 and on page 5 line 21-28.

AC1 reply: We have included this information into Table A1 (Appendix).

RC1 specific comment 9:
The use of brackets rather than commas can be a matter of personal preference, but in Section 2.1 this results in parts of the text being awkward to read. A particular example to address is on Page 4, Line 16 where nested brackets are used, but are incomplete. To aid the reader, I'd suggest that this and other sentences be reworded to allow many of the brackets to be removed.

AC1 reply: Thank you for this comment, we have reformulated this sentence and removed brackets to make this more readable, and have read through and tried to remove unnecessary brackets throughout.

RC1 specific comment 10:
Section 3 Page 6, Line 12, reference to Figure 1 - Suggest spiltting Figure 1 left and Figure 1 right into separate figures. This would help enable an explanation/translation of the German text labelling to be included. Unfortunately, the text is inaccessible for those unfamiliar with German.

AC1 reply: We agree and we have now split these figures (now Figs 1,2) and have included the translation of the German text in the caption for (the current) Figure 1 left.

RC1 specific comment 11:
Page 6, Line 13, reference to Figure 2 - It is useful to have Figure 2 included to illustrate game development, but referencing of Figure 2 left (Line 21) and Figure 2 right (Page 7, Line 1) needs to be clarified. For example, it's unclear if there should be a reference to Figure 2 middle and if so, it's very unclear what Figure 2 left actually illustrates and what it adds to the documentation of game development.

AC1 reply: Thank you for noticing this. The figure numbers are incorrect in the text and the references should be to Figure 1 left and right and not Figure 2 left and right. We have now separated these figures to make the text clearer (now Figs 2, 3).

RC1 specific comment 12:
Page 6, Line 25, reference to Figure 3 - Including an explanation/translation of the German labelling would help understanding of the Wiapuna game.

AC1 reply: We agree have now included a translation of the German text in the figure (new Fig 3).

RC1 specific comment 13:
Page 7, Line 12, reference to Table 1 - Column headers include Price/year and Yield/year, but the units for price and yield are no specified. If the intention is that they are dimensionless and illustrative in the context of the game, then this needs to be clarified.

AC1 reply: These values are given in arbitrary monetary units, and have added this to the caption in Table 2.

RC1 specific comment 14:
Section 3.1, Evaluation of learning outcomes The key messages from game development seem to relate mainly to insufficient time, planning challenges and need for re-timetabling of other course modules. This is interesting, but the evaluation would benefit from documenting more substance on the value and benefits to learning about water resource sharing derived from the games developed. In this context, the conclusions on the game development state that the "students had to think through the intricacies and complexity of water resource sharing, as they thought through players' moves and water resource outcomes", but there is no detail on what these intricacies and complexity were. This is in contrast to the learning experiences from using Irrigana noted in Section 2.1, which at least highlights that the learning has been that "cooperative behavior and communication were both key to succeeding".

AC1 reply: Thank you for this helpful comment. We have now added more text into the Discussion and Conclusions section to elaborate upon the learning outcomes from game development, and to tie together with learning outcomes from game play (see also reply to RC1 comment 4). You can find the added text on pages 10-11, with relevant references.

RC1 specific comment 15:
It would improve the paper's contribution if it identified the specific points of learning on water resource sharing that have been derived from developing the games.

AC1 reply: Yes, we agree and have added more discussion on this (please see replies to comments 4, 14).

RC1 technical correction 16:
Page 1, Line 23 - Reference to Johnson, 2012 should either be Johnson et al. or the paper is missing from the reference list.

AC1 reply: Yes, this is incorrect and should refer to Johnson et al. (2012). We have corrected this in the text.

RC1 technical correction 17:

Page 2, Line 20 - To improve clarity, suggest rewording as follows, ".... role of cooperation in, and competition for the use of water as a limited common-pool resource"

AC1 reply:  Thank you, this has been reworded.

RC1 technical correction 18:
Page 3, Line 15 - Reference should read Lecoutere et al. (2015)

AC1 reply:  Thank you for noticing this, it has been corrected.

RC1 technical correction 19:
Page 5, Line 21 - Rewording suggested as follows "Additional analysis was carried out considering user data collected since July 2013, when user histories began to be saved; this excluded data collected during our own use of Irrigana. This was done to further analyse how ......"

AC1 reply:  We have reworded the sentence for clarity.

RC1 technical correction 20:
Page 7, Line 7 - Insert "a" as follows, ".....Heins (1994), as a way to show.."

AC1 reply:  Thank you for noticing, we have corrected this.

We would like to thank Michael Jones for his very careful review of our paper and his valuable comments.  By including his more substantial comments on the irrigania questionnaire, and further clarifying the learning outcomes, we feel that our manuscript has been substantially improved.

**Reply to reviewer 2 on "Learning about water resource sharing through game play" by T. Ewen and J. Seibert**

**Reply to RC2: Anonymous Referee #2**

RC2 comments:
The paper presents an interesting and innovative learning tool to understand resource management and use. The manuscript begins with a review of a range of games available but no critical input is provided as to what the limitations are of the reviewed examples and why the new game presented is different. No important contribution is put forward as to 'what is the new aspect this new game provides that hasn't been provided already by the other games?' the review is therefore short of analytical substance and would require more work in order to identify gaps in the current knowledge and use of these types of games and how the new game presented is different and ultimately better?

AC2 reply:  We thank the reviewer for this helpful comment.  We have now included more literature in the introduction to help identify the gaps in the current literature regarding other types of games that are currently used for teaching about water resource sharing. We hope this helps to better compare Irrigania with the other games, and allow the strengths of Irrigania to be better identified.

RC2 comment:
The manuscript lacks a proper discussion of the implications of the use and results of the game once it has been played.

AC2 reply:  This is a very helpful comment.  In the text we wrote that "cooperative behavior and communication were both key to succeeding", which was actually based on feedback from teachers who had discussed the outcomes with their classes after the students played.  In some cases, students played on more than one occasion, and usually students notice that these factors (cooperative behavior and communication) are key to succeeding and so approach the next game with this in mind (and thus usually change their strategy based on this outcome).  We have now tried to make these "implications of the use and results of the game" more clear in the text, and have added text to the Discussion and Conclusions section on pages 10 and 11.

RC2 comment:
The manuscript should include a section on implications for management, and a discussion as to how these results are relevant in the real world?

AC2 reply:  Thank you for this helpful comment. We have now included some text (see reply to RC2 comment above), supported by relevant literature, which helps to support how the learning outcomes from both playing and developing games might be carried over from the classroom into the workplace.

RC2 comment:
How can managers/practitioners learn from this new knowledge and advance groundwater management? What should be the lessons and messages to take home with that?

AC2 reply:
Although we refer to the fact that Irrigania may be useful for water resource managers, we don't currently have any feedback from this user group to (we feel) support any further comments on this.  We have however commented on this in the text as regards to student learning in the classroom, and how this learning in the classroom setting may be relevant for these students in further careers in water management (see replies to RC2 above).  The new text can be found on pages 10,11.

RC2 comment:
The scope of the manuscript is therefore limited to the 'classroom' and doesn't do much to advance 'further and wider knowledge' on groundwater management. The manuscript therefore lacks 'vision' and would require rethinking as to the real lessons to be drawn from the work that is presented.

AC2 reply:  Although the scope of our manuscript is indeed clearly focused on "classroom" aspects, we believe that learning about groudwater management starts in the classroom -- it is in the classroom where future water resource managers are trained, and think that this learning does get carried forward.  It would be nice to have some feedback/data from water resource managers and practitioners to further identify real lessons.  Although our data is currently limited to teaching about water resource sharing in the classroom, we strongly believe that there is value in this information to better improve our educational programs and training in water resource management.  We do however agree with this comment insofar as we could try to connect our findings with how they might feed into real world lessons.  We have added some text to address this in the discussion (pages 10,11, see also replies above).

RC2 comment:
Further details on the data used (as suggested by the other reviewer) in the form of a table with descriptive statistics of the results would be interesting to have.

AC2 reply:  Thank you for this comment.  We agree and have added a summary table of the survey (Table A1), also according to RC1's comments (and outlined in replies to RC1 comments 3, 8).  We hope this will help to clarify and better explain the results, and improve the readability of the manuscript.

We would like to thank reviewer #2 for all the helpful comments and questions. Although we would like to be able to better address the questions related to "vison" and real lessons in water resource management, our current study (and data) is limited to the classroom.  We have however tried to add discussion into the final section to address these points, as they are relevant and would be very interesting to pursue as a follow-up to this study.

**Summary list of all relevant changes to the manuscript:**

- Abstract: added findings and results, lines 16,17, 18-20
- Section 1: corrected citation Johnson et al., 2012 page 1, line 25
- Section 1, page 2, lines 6-15, 18-35: added a more critical appraisal of current literature on game play for water resource sharing (with relevant literature), to make clearer in what ways Irrigania is a good tool for teaching about water resource sharing and advances this.  Revision to support comments by the editor (Iain Stewart) and RC2.
- Section 2, page 3, lines 24-27: added description of cost/revenue to irrigate with groundwater/rain water as revision to RC1 comment 5.
- Section 2, page 3, line 29: explanation of income (net of farmer revenue and costs), RC1 comment 6
- page 4, lines 4-14; 19-27, text added to explain two scenarios of game play added in Table 1, RC1 comment 2
- page 5, line 2, added brackets to Lecoutere et al. (2015) ref, RC1 technical correction.
- page 5, line 11, added reference to Table A1, summary of survey results, RC1&RC2 comments
- page 7, lines 6-7 reworded sentence for clarity, RC1 comment technical correction
- page 8, line 35, removed German text as it's now added in Table/Figure as English translation
- Section title 3.1, "Evaluation of learning outcomes" changed to "Evaluation of "Water Games" course.  Discussion of learning outcomes has been added to Section 4, so we have changed the title to better reflect the text in this section.
- page 10, line 26-page 11 line 4, continued page 11 lines 14-24, text added with discussion of learning outcomes of playing and developing games, and broader reaching effects of taking lessons from the classroom to the workplace, RC1 comments 4,14, editors comments.
- page 11, last sentence, removed as it no longer fit with new text.
- modified Figure/Table numbers accordingly in main text
- references added: Burton 1989; Burton 1994; Corrigan et al., 2015; Hummel et al., 2010;  Jones 2011; Madani 2010; Magombeyi et al., 2008; Medema et al., 2016; Ruben 1999; Wouters et al., 2009.
- Figure 1 added, to show the interface of Irrigania game play, RC1 comment 2
- Figure 2, 3 now split, RC1 comment
- Figure 3, have added English translation of German text to figure
- Figure 4, added English translation of German text on boards to caption
- Figure 5, added English translation of German
- Table 1 added to show two Irrigania scenarios
- Table 2, added "units" for price, RC1 comment
- Appendix, Table A1 added with summary of Irrigania survey results, RC1, RC2 comments

[revised manuscript text omitted]